# Estimating the Impact of Urban Planning Concepts on Reducing the Urban Sprawl of Ulaanbaatar City Using Certain Spatial Indicators

**Bolormaa Batsuuri [1,*], Christine Fürst [2] and Buyandelger Myagmarsuren [3]**

[1]   Department of Geography, School of Arts and Sciences, National University of Mongolia, Ulaanbaatar 14200, Mongolia

[2]   Institute for Geosciences and Geography, Department for Sustainable Landscape Development, Martin Luther University Halle-Wittenberg, 06108 Halle, Germany; christine.fuerst@geo.uni-halle.de

[3]   Department of Land Management, School of Agroecology, Mongolian University of Life Sciences, Ulaanbaatar 17024, Mongolia; buyandelger.m@muls.edu.mn

*   Correspondence: b.bolormaa@num.edu.mn

**Abstract:** The urban sprawl process of Ulaanbaatar has changed dramatically due to population growth. Ulaanbaatar city land management master plan defined the settlement zone area suitable for living as 33,698 ha. However, due to unrestricted urban sprawl caused by the exponential growth of the city's population, the settlement zone area reached 39,235 ha, which exceeds the limit by 5537 ha. In order to tackle this issue, several urban planning concepts were developed to be implemented within Ulaanbaatar city urban planning framework. It is, in any case, problematic to choose a single planning concept due to the fact that neither measurements nor analyses are being made of the respective spatial indicators in urban planning assumptions that are taking urban form into consideration. One of the prerequisites for identifying an optimal concept in urban planning is an assessment of urban form, and measuring the impacts against its spatial data. This study uses 1990–2020 satellite image data to investigate the urban form of Ulaanbaatar with a future action plan. Using remote sensing and GIS technology, Ulaanbaatar city sprawl was analyzed for defining urban form, and consequent results were obtained by comparatively measuring the impacts of monocentric, polycentric, and compact city concepts on city sprawl by applying spatial indicators that have been used in the world's major cities. The study results show that the compact city concept is the optimal solution to reduce uncontrolled city sprawl based on a technical point of view. This will lower Ulaanbaatar's sprawl threefold and compress the urban settlement area down from 39,235 ha to 12,479 ha.

**Keywords:** land-use form; land efficiency; spatial analysis; GIS; Ulaanbaatar; Mongolia

---

## 1. Introduction

The urban sprawl of urban areas from the earliest stages until now is the result of several internal and external factors [1]. One of the most serious problems in the 21st century is global population growth and its consequent urban sprawl, particularly in developing countries [2]. This trend, which is observed in developing countries, has emerged in Ulaanbaatar city since the 1990s in relation to the country's shift from a socialist to a market-oriented economy [3]. The growth is mainly attributed to the large influx of rural to urban migration. The aggregation of the population in the city has caused many problems, including informal settlement air pollution and a lack of affordable housing. Inhabitants fear the impacts of informal settlement and migrant populations on neighborhood

character and the extra pressure on local services [4]. Urbanization was studied in relation to economic development and technology by Paddison and Hutton, Corey, Wilson, and Pan; city rescaling was investigated by Harrison; city inclusivity was researched by Pieterse; environmental sustainability was studied by Whitehead; theories of place were investigated by Ho; labor markets were researched by MacKinnon; redundant and marginalized Spaces were studied by Turok; and Urban Governance was investigated by Deas and Headlam [4]. Since the 1970s, Geographic information systems (GISs) have emerged in urbanization studies, providing new or previously barely applicable techniques for handling and analyzing spatial data [5]. The GIS provides many tools for handling all issues related to land resource management. One of the key characteristics of GIS technology is that it is capable of handling and combining different types of data very efficiently [5,6]. Most of the developed countries are well equipped and updated with a detailed spatial database, lack of spatial data persists in developing nations [7]. Therefore, for the developing countries, remote sensing proved its effectiveness for spatial data updating [8]. For instance, a significant part of urban planning in Ulaanbaatar city ignores remote sensing and GIS technology. For that reason, it is complicated to measure and compare the impact of future urban development action plans and analyze its structure. In cities such as Houston in the USA, Shanghai, and Nanjing in China, the spatial indicators based on the usage of remotely sensed data and GIS spatial modeling were used to assess the current conditions of urban form and apply the results for future urban planning are becoming of interest. Therefore, the sprawl process of Ulaanbaatar city needs to be analyzed using spatial indicators, and the results should be applied for urban planning purposes to define urban form. Adolphson explained urban forms nine main categories, as shown in Figure 1 [1].

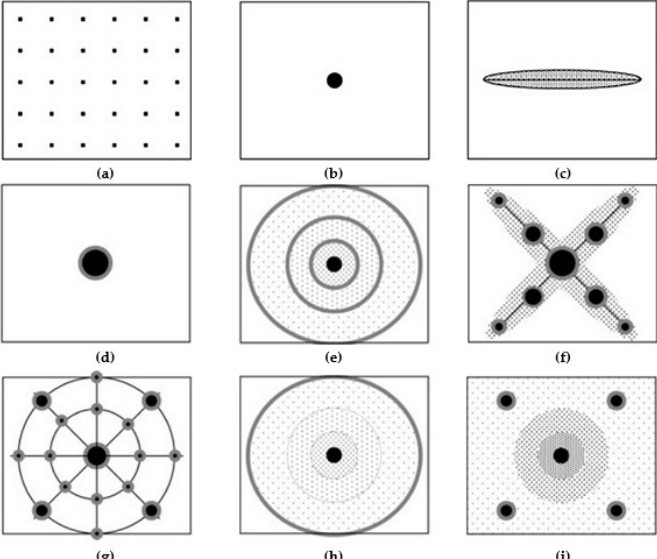

**Figure 1.** Various urban forms. (**a**) areal urban structure; (**b**) point urban structure; (**c**) linear urban structure; (**d**) compact urban structure; (**e**) dispersed urban structure; (**f**) corridor urban structure; (**g**) multinucleated urban structure; (**h**) fringe urban structure; (**i**) ultra-urban structure.

The general scope of this paper is to measure the urban form and impacts of its concept, at the same time to compare impacts of urban concepts based on land-use form and land-use efficiency and determine the optimal concept for reducing urban sprawl of Ulaanbaatar city from a technical point of view using a mathematical procedure and GIS urban analysis. The research outcome demonstrates the current urban form of Ulaanbaatar city and presents the urban sprawl in six times series, from 1990 to 2020 based on satellite data using GIS. As of today, urban planning in Ulaanbaatar City has involved a combination of monocentric, polycentric, and compact city concepts, but, for future planning, one optimal concept that is reducing urban sprawl should be identified based on a technical point of view. To the best of our knowledge, no scientific studies have been reported on planning

concepts in Mongolia in which the technical point of view of land-use form and efficiency are used. Therefore, the purpose of the research at hand is to estimate and analyze the impact of urban planning concepts from a technical point of view by identifying Ulaanbaatar City's uncontrolled sprawl using spatial indicators and comparisons of land-use form and land-use efficiency quantitative data. The research has the following objectives:

1. To study the current situation of Ulaanbaatar City's uncontrolled sprawl starting in 1990 and identify its coverage and define current urban form;
2. To analyze the impacts of different urban development concepts applied in Ulaanbaatar City's master plan on the urban form in relation to urban sprawl by using spatial indicators; and
3. To estimate the impact of each urban development concepts by comparing and integrating using certain spatial indicators on reducing the urban sprawl of Ulaanbaatar city.

## 2. Methods and Data

### 2.1. Study Area

To achieve the aims of the research, we analyzed the urban sprawl of Ulaanbaatar, the country's capital, which has been one of its fastest-growing cities over the past few decades as an example (Figure 2). Ulaanbaatar city has been growing significantly since the mid 20th century and has been considered the country's main urbanized area since then [5]. According to statistical data of Mongolia, Ulaanbaatar's population grew from 0.6 million in 1990 to more than 1.53 million in 2020, an increase of 2.5 times. The first scientifically-based urban development master plan of Ulaanbaatar city was developed in 1954 at the "Gypragor" Institute in Moscow, and it has been revised six times since then. With over a decade of social transition until the Fifth Master plan was developed in 2002, land utilization has been basically managed without any planning concept or general policy [9]. This situation established the foundation for today's uncontrolled urban sprawl [10]. Ulaanbaatar city is located in the center of Mongolia along the southern edge of the Khentii mountain range, surrounded by mountains from four sides of the valley along the river basin. Hence, it has a limited area suitable for settlement. The city's total area is 470,440 ha. The main part of the city is located along the banks of the Tuul River, which originates in the Khentii mountain range at an average elevation of 1350 m above sea level. Most of the residential area stretches out along the northern section of the river. According to the Ulaanbaatar city land management plan, the zone suitable for settlement should be an area of 33,698 ha. This was identified using the following eight factors:

- The slope of the land, with an incline of less than 15°;
- Not in an area exposed to permafrost;
- Outside of forest and river protection zones;
- Not in the National park area;
- Not in croplands;
- Not swampy;
- Not in the zone prone to the risks of flooding or rockfall on the mountain slope;
- Not in an area under engineering limitations or restrictions [11].

Moreover, a total of 15,403 ha of land in the eastern part of Ulaanbaatar was planned for a new settlement zone; however, due to its distance from the city center, this area was not developed.

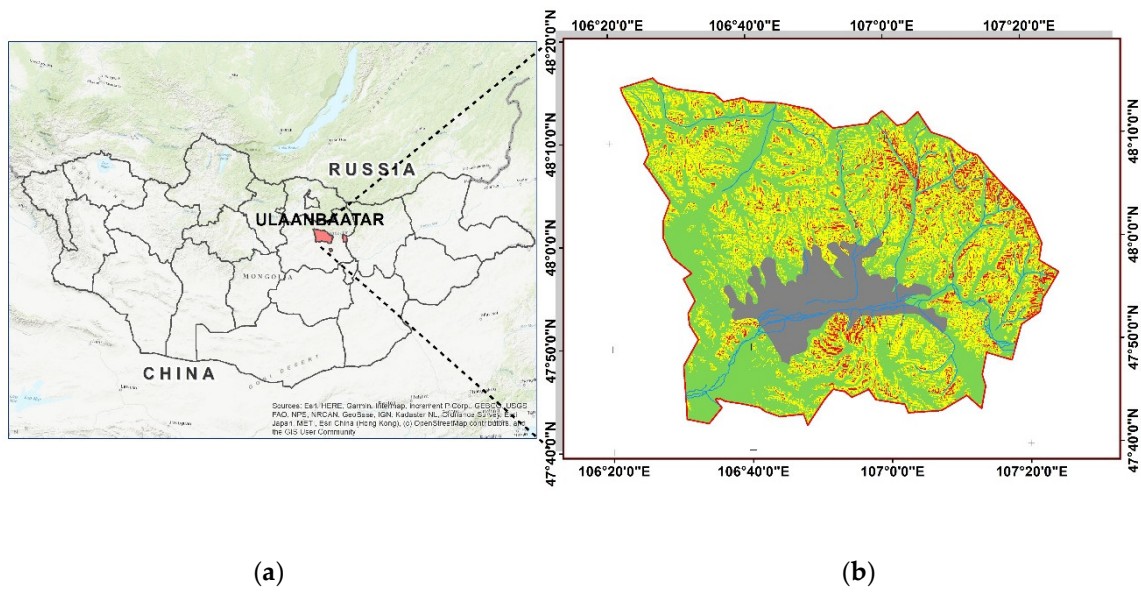

(**a**) (**b**)

**Figure 2.** Map of the study area. (**a**) Location of Ulaanbaatar city; (**b**) area suitable for settlement.

### 2.2. Data Sources and Processing

The current scattered urban sprawl of Ulaanbaatar city needs to be displayed by spatial indicators. Hence, base data for identifying the Ulaanbaatar city settlement zone area were prepared using the geographic information system based on Landsat-1990, Landsat-1995, Landsat-2000, Landsat-2005, Quickbird-2010, World View2-2015, and Sentinel2-2020 satellite data with a frequency of five years covering the period of 1990–2020 (Table 1).

**Table 1.** Data source used to create urban settlement (1990–2020).

| Frequency | Source * | Time Period | Method of Creation |
|---|---|---|---|
| Urban settlement in 1990 | Landsat 5 path/row of 131-027 | 10 September 1990 | Supervised classification |
| Urban settlement in 1995 | Landsat 5 path/row of 131-027 | 7 August 1995 | Supervised classification |
| Urban settlement in 2000 | Landsat 5 path/row of 131-027 | 1 August 2000 | Supervised classification |
| Urban settlement in 2005 | Landsat 5 path/row of 131-027 | 14 May 2005 | Supervised classification |
| Urban settlement in 2010 | Quickbird NW corner Lat = 48.02, SE corner Lat = 47.73, NW corner Long = 106.53, SE corner Long = 107.32 | 20 May 2010 | Digitizing/ georeferencing |
| Urban settlement in 2015 | World View-2 NW corner lat: 48.01, SE corner lat: 47.09, NW corner long: 106.61, SE corner long: 106.72 | 14 September 2015 | Digitizing/ georeferencing |
| Urban settlement in 2020 | Sentinel 2 T48UXU | 7 February 2020 | Digitizing/ georeferencing |

*: For this study, Landsat 5 data downloaded from http://earthexplorer.usgs.gov/ and other data from Monmap Co.ltd.

According to Landsat 5, satellite data containing seven spectral bands with a path/row of 131-027 was obtained and other satellite data were used for the RGB color map. Accuracy assessment is important in land-cover mapping and land-cover change research. Accuracy assessment was performed using data from different spectral bands for the Landsat 5 imagery, but not for the high-spatial-resolution data for 2010–2020 as only RGB color composites were supplied to the research team. Therefore, it was not possible to undergo land coverage classification and accuracy assessment due to a lack of high-resolution spectral bands from the period 2010–2020. So we switched to utilizing the GIS program. Based on the above data, the Ulaanbaatar City sprawl area change was identified using the GIS vector overlay tool. Satellite data were uploaded for each year, and settlement zone boundaries were mapped by digitizing. Then, vector and raster data for each year were overlaid using Arc GIS 10.5 software to identify the area and perimeter for each year. GIS provides many tools for handling all issues related to land resource management.

*2.3. Research Methods*

In order to determine the reasons for Ulaanbaatar city sprawl, analyses of six urban development master plans were made using monograph and deduction methods. Monocentric, polycentric, and compact city concepts were compared with the goal of improving the current land-use form, intensifying land use, and decreasing the scattered urban sprawl in Ulaanbaatar City. Land-use efficiency and land-use form indicators that were used in the cases of Houston, USA and Shanghai and Nanjing, China to measure urban sprawl due to population growth and to analyze and reflect on the results during the urban planning process were selected and applied in this study. The impact of each urban development concept on urban sprawl was comparatively analyzed using six indicators of land-use efficiency [1,12–14] and land-use form [14–16].

2.3.1. Measuring Land-Use Efficiency

As mentioned above, three indicators were selected to identify land-use efficiency. This included the floor area ratio; population density and building coverage ratio. The settlement zone population density was used to determine the building capacity and utilization rate. A low rate of building utilization demonstrates a low population density, whereas increases in building heights and site areas should result in a denser population. The population density in the developed zone serves as a significant factor in decreasing urban sprawl; therefore, it was selected as an indicator. This is expressed in the following formula [1]:

$$d = \frac{P}{A} \tag{1}$$

d—population density;
P—population size;
A—area of urban land use.

The bigger the value of "d" is, the higher the capacity of the development area is [14]. Even if the population density increases, unless the floor and area ratio increases, the building capacity will not improve. In terms of the need to improve the efficiency of land use, the building capacity should be increased. Therefore, the floor and area ratio was selected as a measurement of land-use efficiency:

$$FAR = \frac{\sum_{i=1}^{t} A_i}{S_L} \tag{2}$$

FAR—floor area ratio;
A$i$—site area of urban land use;
t—total floor area;
S$_L$—unit area size.

The higher the value of the construction area floor area ratio is, the bigger the capacity of the building is. This means that land use is more efficient [1,12]. A city expands not only outwards but also inwards due to the building density. When the building coverage ratio increases due to land-use

efficiency, it causes sprawl in the urban development area. Hence, the city's inner form is lost, leading to negative impacts such as reductions in public use areas, roads, and green areas. The process of urban inward sprawl caused by an increased building area is defined by the building coverage ratio [1,13]:

$$BCR = \sum \frac{F}{A} \tag{3}$$

BCR—building coverage ratio;
F—building area;
A—area of urban land use.

### 2.3.2. Measuring Change in Land-Use Form

Dimensions of land-use form change can be easily shown on a map and in areal and satellite images. Hence, comparative analyses were made using the following three indicators (i): the fractal dimension, compactness index, and Feret's diameter. There are several methods that can be used to measure the fractal dimension including "self-similarity", "divider", "Hausdorff", "correlation", "box counting dimension", "dilation dimension" and others [17,18]. The fractal dimension was calculated as mentioned above using a map and aerial and satellite imagery. Hence, the "box counting dimension" was used in this research. This method was first identified by Mandelbrot [19] and was used to detect fractions of complex forms, such as urban morphology [20]. Fractal dimension values range between 1 and 2, and if the forms are more complex, the fractal dimension value approaches 2 [21]. The following formula was employed for the calculation of each land-use fractal dimension [22]:

$$D = \frac{2 \log P/4}{\log A} \tag{4}$$

D—fractal dimension;
P—external perimeter of urban land-use area;
A—area of urban land use.

The Feret's diameter is calculated with the following formula using the maximum distance between a pair of coordinates [23] and the maximum distances covering each land use [24]:

$$F = max_{i,i+1(d_{i,i+1})} \tag{5}$$

F—Feret's diameter;
*i,d*—land-use end coordinates.

The Feret's diameter presents the land-use form ratio within the F ≤ 1 value range. When F = 1, the shape perimeter resembles a square; if the F value approaches 0, the shape form ratio is lost and tends to stretch to one side. The shape ratio in urban geographic research is the key indicator for urban sprawl. Therefore, a change in the shape ratio demonstrates urban sprawl in the following way: if the city expands outwards, the shape ratio decreases, whereas an increased internal density results in an increased shape ratio [25]. In 1982, Ritter et al. [26] attempted to identify a simple ratio by comparing the shape perimeter to its area when defining the shape density, the basic method for which was developed by Richardson [27]:

$$C = \frac{2\sqrt{\pi A}}{P} \tag{6}$$

*C*—compactness of urban land-use area;
*P*—total land-use perimeter;
*A*—area of urban land use.

## 3. Results

### 3.1. Spatial Analysis of Urban Sprawl of Ulaanbaatar City

Landsat satellite data from 1990, 1995, 2000, and 2005, Quickbird data from 2010, World View data from 2015, and Sentinel2-2020 satellite data from 2020 from Ulaanbaatar were used to determine the extent of urban sprawl. Sprawl increased by 540 ha in 1990–1995, 3871 ha in 1995–2000, 2185 ha in 2000–2005, 9285 ha in 2005–2010, 9484 ha in 2010–2015, and 3210 ha in 2015–2020 (Figure 3).

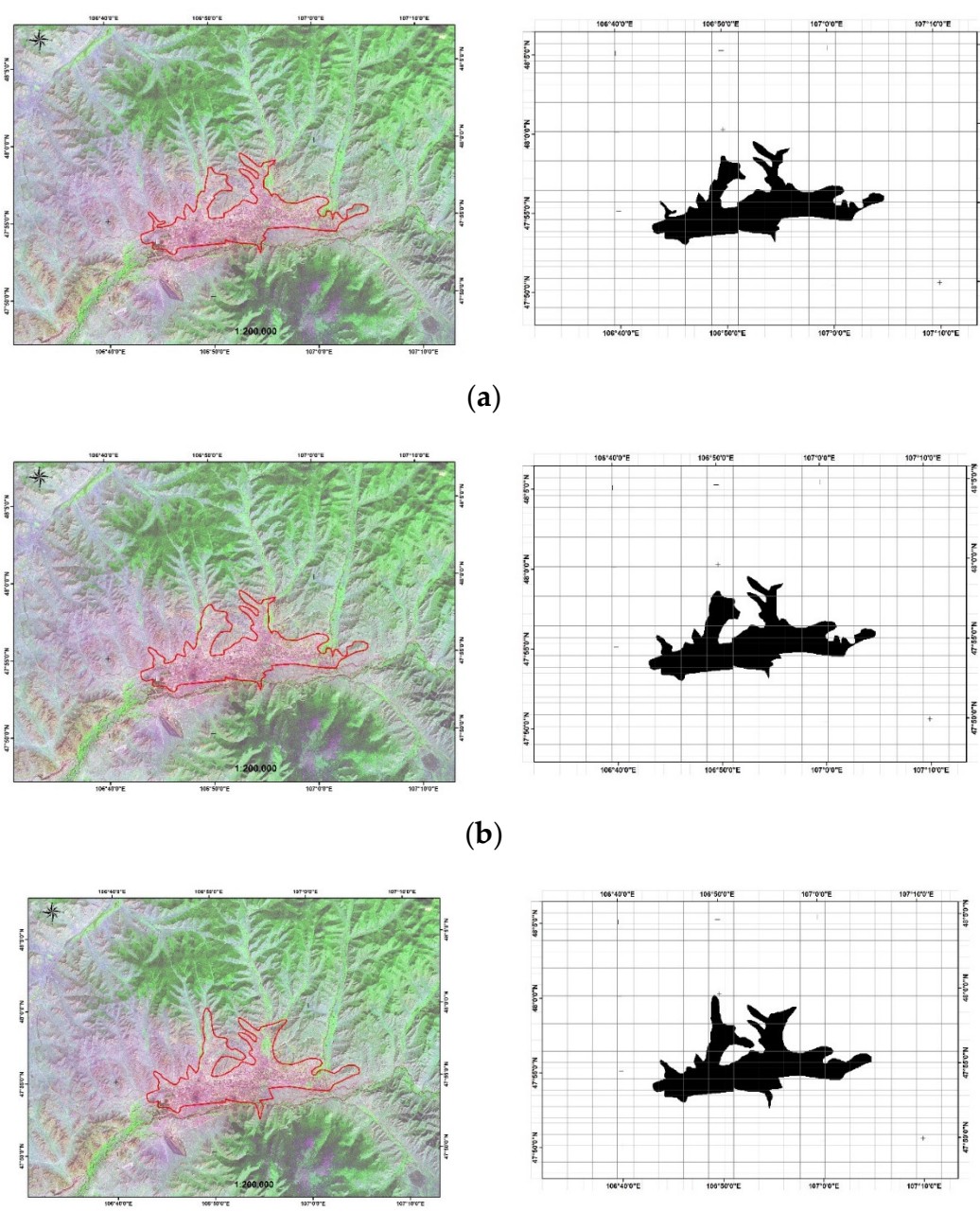

(**a**)

(**b**)

(**c**)

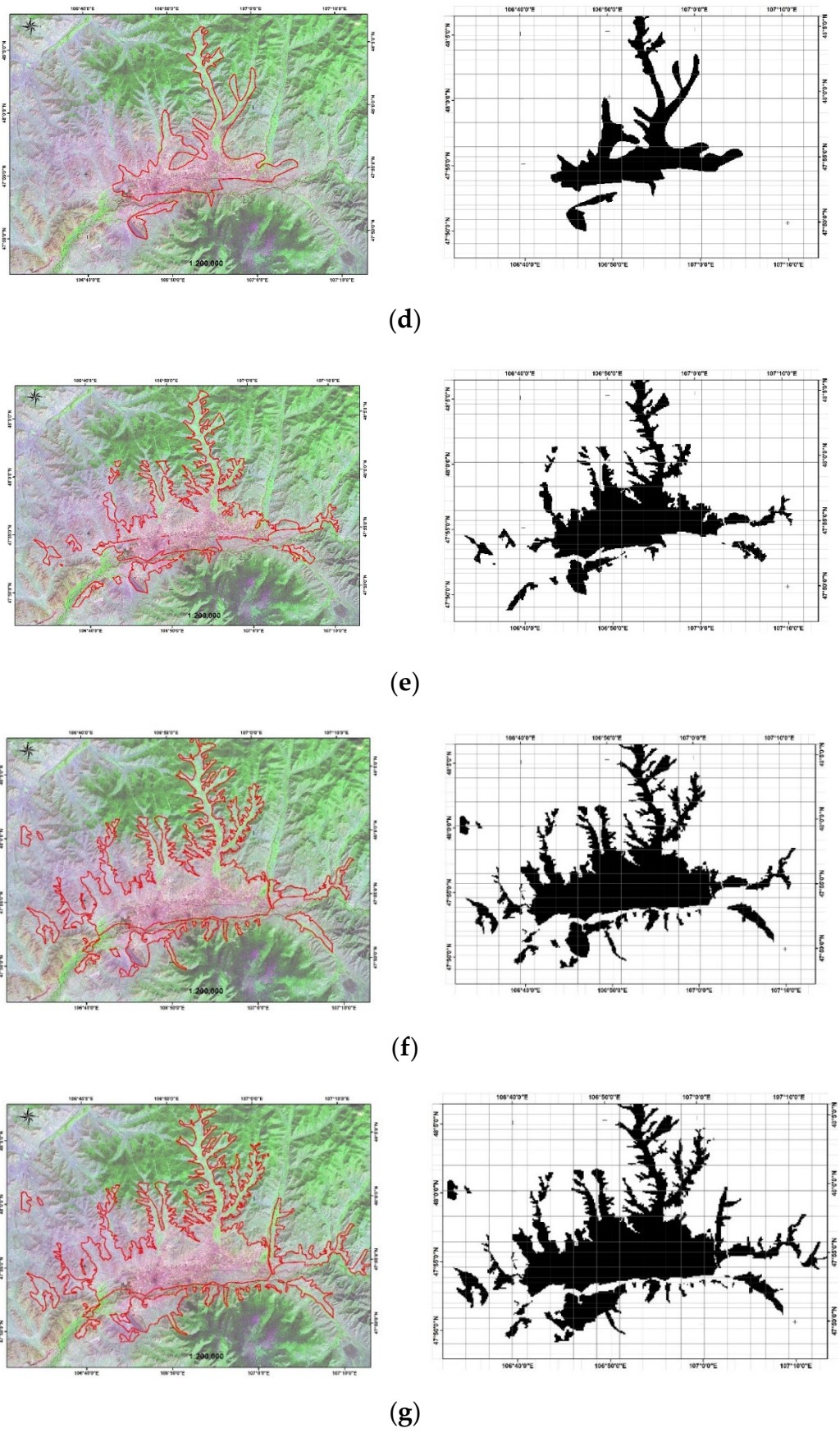

**Figure 3.** Map of the urban sprawl showing increases of (**a**) 10,730 ha by 1990; (**b**) 11,270 ha by 1995; (**c**) 15,141 ha by 2000; (**d**) 17,326 ha by 2005; (**e**) 26,541 ha by 2010; (**f**) 36,025 ha by 2015; and (**g**) 39,235 ha by 2020.

Based on the above data, the following results were used to determine the spatial changes in land use (Figure 4).

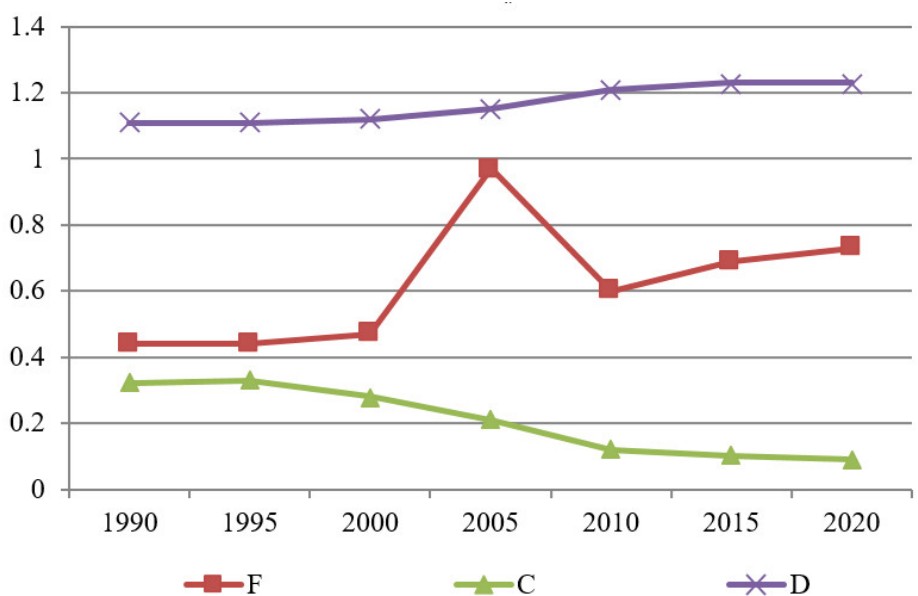

**Figure 4.** Changes in the land-use form of Ulaanbaatar city (1990–2020).

According to Table 2, Feret's ratio was 0.44 in 1990, indicating that Ulaanbaatar had a corridor urban structure, and it was 0.73 in 2020, indicating that the sprawl of land-use form has expanded mostly vertically over the last 20 years. However, considering the value of this ratio every five years, in 1990–1995 and 1995–2000, the ratio expanded horizontally (between 0.44 and 0.47), and between 2000 and 2005 the ratio increased to 0.97. Thus, the urban form expanded vertically, and the structure of the city became a relatively standard and compact urban structure. In 2005–2010, the ratio expanded to 0.60 in the horizontal direction, while in 2010–2015, it increased to 0.69, and in 2015–2020, it expanded to 0.73 back in the vertical direction. The compactness ratio of Ulaanbaatar has been steadily declining from 0.28 to 0.09 over the past 15 years, indicating that Ulaanbaatar has expanded rapidly with a low urban density. The fractal dimension rate was 1.23, which fits into the second level of the four-level [28] classification from the School of Geography at the University of Florida in the United States. As of today, the results of the above three spatial indicators of land-use form spatial analysis present that Ulaanbaatar city is expanding in a dispersed urban structure of nine main categories. The Ulaanbaatar City land management master plan defined the settlement zone area suitable for living as 33,698 ha. According to Table 2, Ulaanbaatar city's population, the settlement zone area reached 39,235 ha, which exceeds the limit by 5537 ha. There are 111,086 inhabitants living in 5537 ha of natural hazards area.

**Table 2.** Changes in the land-use form of Ulaanbaatar city (1990–2020).

| Он | Area/ha/ | Perimeter | Feret's Diameter /Max/ | Feret's Diameter /Min/ | Feret's Diameter (F) | Compactness (C) | Fractal Dimension (D) |
|------|----------|-----------|--------|--------|------|------|------|
| 1990 | 10,370 | 112,600 | 27,321 | 12,183 | 0.44 | 0.32 | 1.11 |
| 1995 | 11,270 | 115,300 | 27,606 | 12,184 | 0.44 | 0.33 | 1.11 |
| 2000 | 15,141 | 155,927 | 25,955 | 12,155 | 0.47 | 0.28 | 1.12 |
| 2005 | 17,326 | 220,076 | 27,270 | 26,390 | 0.97 | 0.21 | 1.15 |
| 2010 | 26,541 | 483,850 | 45,880 | 27,580 | 0.60 | 0.12 | 1.21 |
| 2015 | 36,025 | 697,519 | 48,370 | 33,140 | 0.69 | 0.1 | 1.23 |
| 2020 | 39,235 | 772,190 | 50,321 | 36,932 | 0.73 | 0.09 | 1.23 |

*3.2. Comparison of Urban Planning Concepts*

3.2.1. Analysis Using the Monocentric Concept in Ulaanbaatar City

The current basic model of the urban spatial structure of the monocentric city that we are utilizing in modern times has been created by Alonso [29]. According to this traditional model, the city priority aim is to become monocentric, which allows all the activities within a city to take place in the central business district areas such as downtown [30].

The first urban master plan for Ulaanbaatar based on the monocentric concept, which has become the foundation of its development, was approved in 1954, and the population of the city was expected to reach 125,000 over the next 20 years. The second master plan was developed and approved in 1961 and was scheduled for another 20 years, predicting a population of 250,000. The plan was implemented over a relatively slow period compared with the original plan of 14 years, during which time Ulaanbaatar expanded along the Tuul River valley to a length of 20 km and a width of 6–8 km with a built-in area of 3900 ha. As the city's population reached 348.7 thousand in 1975, a third master plan was developed and 19 apartment districts were planned, increasing the housing stock by 79% compared to that in 1960 [10]. During the implementation of this plan, Ulaanbaatar regained its current appearance, but the main drawback was the miscalculation of population growth, as in the previous plan (Figure 5). During the implementation of the third master plan, the population was expected to increase by 50–80 thousand, but in reality, the capital city's population doubled to 492.2 thousand. Hence, it was necessary to update the master plan ahead of time.

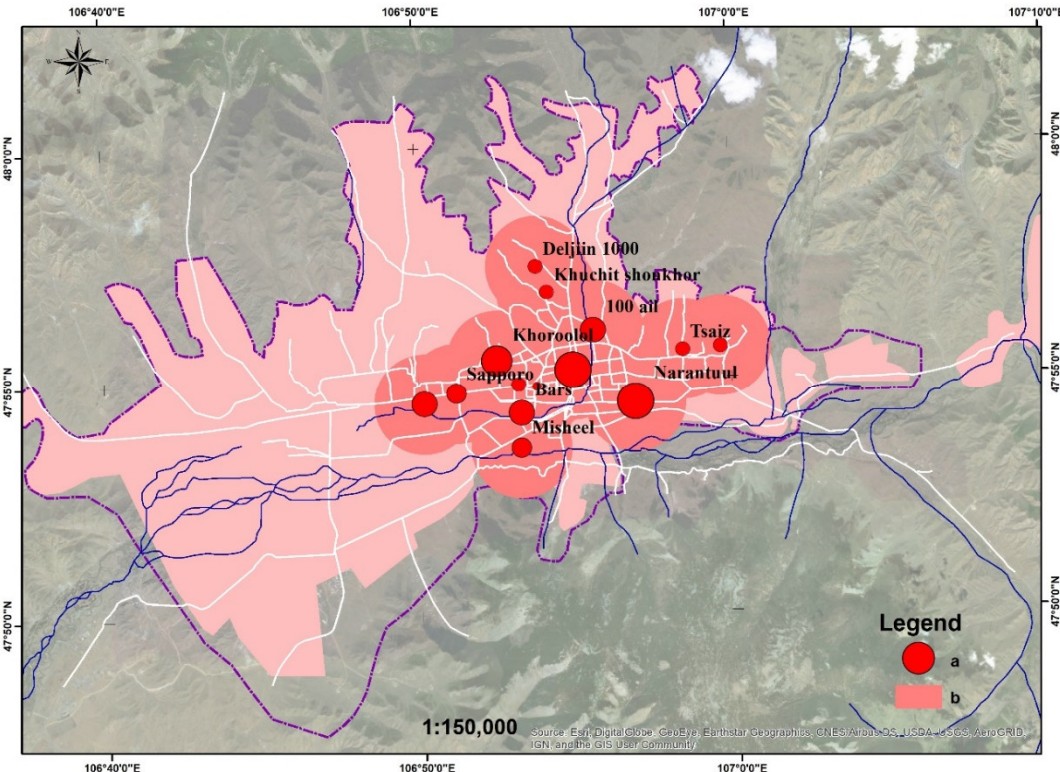

**Figure 5.** Map of the monocentric concept. (**a**) Population concentration points; (**b**) area suitable for settlement.

Land-Use Efficiency Analysis

The following results were obtained from a random sample of 19 districts in the residential area that were created within the framework of the master plan. An analysis of the land-use efficiency was conducted (Table 3).

**Table 3.** Land efficiency of the monocentric concept.

| No. | Indicator | Unit of Measurement | Before the Project |
|:---:|:---:|:---:|:---:|
| 1 | Building coverage ratio | % | 33.7 |
| 2 | Population density | People per ha | 65.6 |
| 2 | Floor area ratio | Unit | 1.6 |

Table 3 shows that the building coverage ratio in the residential area formed during the monocentric concept was 33.7%, which is a low-density residential area. The floor area ratio was 1.6, which means that there were a lot of low-rise buildings, and the population density was 65.6 people per ha, which indicates a small number of residents. In other words, low-rise residential buildings of 1–5 floors were built covering large distances in this area. The results of the study suggest that the monocentric concept favors low-density urbanization over a relatively large area. Therefore, 65.6 people per ha, which is a relatively small number of people, live in an area of 17,326 ha, which represents over half of Ulaanbaatar's suitability zone.

Land-Use Form Analysis

According to the analysis of the monocentric concept, the general plan solution developed during the concept was changed in 1986, but due to the social transition period that occurred until 2010, the quantitative value from 2010 was used to measure the sprawl.



Since Ulaanbaatar is located along a river valley, the sprawl spread horizontally depending on the main road network and the distance to the center, as shown by Feret's aspect ratio (0.60) from Table 4. The low compactness index of 0.12 indicates outward sprawl, and the fractal dimension of 1.21 indicates a clustered form. In general, the sprawl formed by this monocentric concept resulted in transverse, chaotic, and low-density settlement.

**Table 4.** Fractal dimension under the monocentric concept (MC).

| Term | Area/ha/ | Perimeter | Feret's Diameter /Max/ | Feret's Diameter /Min/ | Feret's Diameter (F) | Compactness (C) | Fractal Dimension (D) |
|------|----------|-----------|------------------------|------------------------|----------------------|-----------------|-----------------------|
| MC | 26541 | 483850 | 45,880 | 27,580 | 0.60 | 0.12 | 1.21 |

3.2.2. Analysis Using the Polycentric Concept in Ulaanbaatar City

The first model of a polycentric city was developed by Fujita and Ogawa [31]. The main theory of this model focuses on the spatial distribution of cities while abstracting from the intra-city spatial structure [30]. The Ulaanbaatar city's population reached 492.2 thousand in 1986, and the fourth master plan was developed to modify the third master plan based on a polycentric model. Approved in 1986, the master plan proposed a polycentric approach to decentralization and relocation to suburban areas through a system of group settlements, but this failed in 1990 due to a change of government and an economic crisis and has been forgotten for over 10 years. During this period, land-use forms in the capital city continued to be largely unregulated and chaotic, requiring more evaluation of its future planning and management and the identification of land-use trends [9]. In line with this requirement, the Fifth Master Plan, the plan for the development of the capital city of Ulaanbaatar until 2020, was developed and approved in 2002. Although this plan is unique in that it was the first of its kind as Mongolians developed the plan by themselves, it is essentially a continuation of the polycentric concept of the Fourth Master Plan, developed in 1986. The main content of the master plan is to create an optimal territorial system for the development of the capital's satellite settlements in order to decentralize the urban population. During the planning period, Ulaanbaatar was considered to be a self-sustaining and competitive territorial and economic complex with 15 satellite towns and villages and four zones. Although 10 years have passed since the implementation of the Master Plan, the development of peri-urban towns and villages has remained stagnant, centralized, and dependent on Ulaanbaatar, which has made it impossible to control the flow of migrants, resulting in urban sprawl from Ulaanbaatar city. Adverse effects such as air pollution, traffic congestion, and soil pollution have increased. This situation indicated that the implementation of the plan was insufficient and needed to be clarified [10]. Thus, in 2012, the Sixth Master Plan for the development of Ulaanbaatar until 2025 was developed, which estimates that the population of Ulaanbaatar will grow to 1.4 million. In order to avoid the current centralized system of the city, it has become clear that the development of satellite villages cannot be implemented, so there is a plan to develop the city's internal architectural space into eight sub-centers to partially solve problems faced by the city (Figure 6).

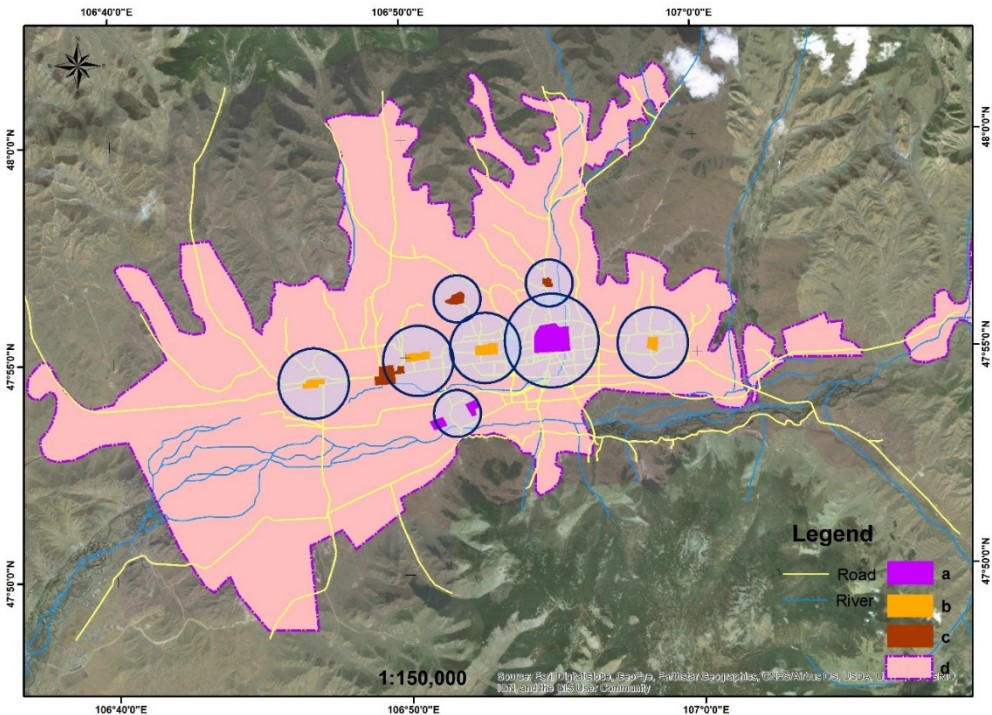

**Figure 6.** Planned polycentric concept. (**a**) city center; (**b**) district center; (**c**) planned sub-centers, (**d**) area suitable for settlement.

Each sub-center will have its own complex of administrative, trade, services, cultural, educational, sports, and social infrastructure service centers and will be responsible for providing social infrastructure to the unrestricted settlement areas created by urban sprawl (Figure 7).

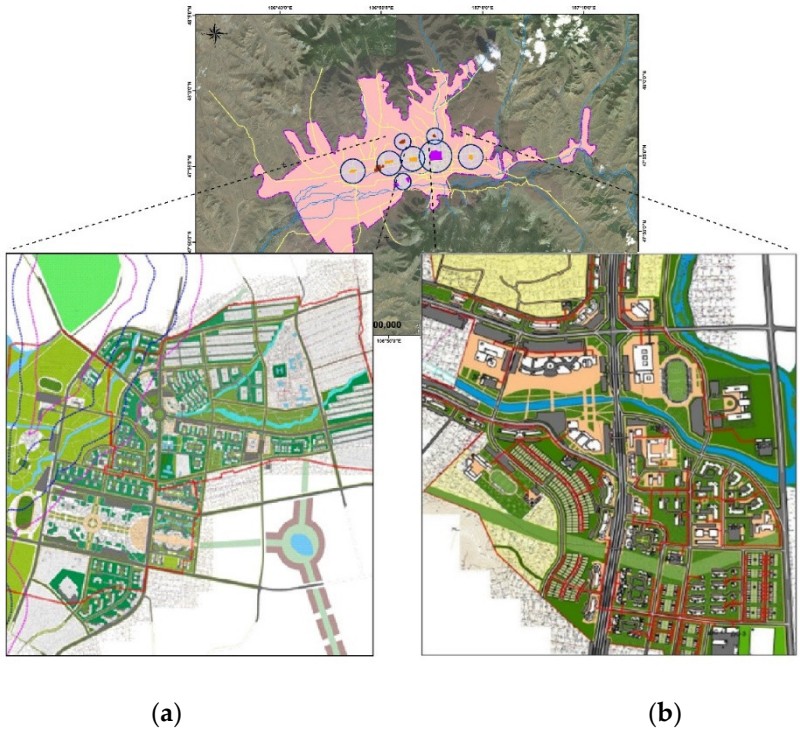

(**a**)                                          (**b**)

**Figure 7.** Planned sub-centers. (**a**) Bayankhoshuu sub-center; (**b**) Selbe sub-center.

Currently, four years have passed since the establishment of the Selbe and Bayankhoshuu sub-centers in accordance with the plan, and other social infrastructure works are presently underway.

Land-Use Efficiency Analysis

As mentioned above, the master plan will be implemented over nine years, so we analyzed the expected results of the plan (Table 5).

**Table 5.** Land efficiency of the polycentric concept.

| No | Indicator | Unit of Measurement | Before the Project |
|----|-----------|---------------------|--------------------|
| 1 | Building coverage ratio | % | 43.7 |
| 2 | Population density | People per ha | 50 |
| 2 | Floor area ratio | Unit | 1.8 |

Table 5 shows that the building coverage ratio in the residential area planned in the polycentric concept is 43.7%, which represents a low-density residential area. The floor area ratio is predicted to be 1.8, which indicates that there will be a lot of low-rise buildings, and the population density is predicted to be 50 people per ha, which points to a small number of residents. Although the building coverage ratio and floor area ratio is predicted to increase, the predicted population density is lower than that for the monocentric concept city. This is due to the fact that the two sub-centers are responsible for providing social infrastructure for the entire unrestricted settlement area in the northern region.

Land-Use Form Analysis

In terms of the polycentric concept, the quantitative values from 2020 were used to measure the urban sprawl scale, as they are the main outcome of the current master plan. In a polycentric development concept, it is planned that the sub-centers will be expanded vertically in terms of urban land-use form with a Feret's aspect ratio of 0.73 and a compactness ratio of 0.09 in order to provide social infrastructure to reduce the urban sprawl. Due to this process, the fractal dimension, which is an indicator of the urban growth dynamics will reach 1.23, indicating that the form of the city will become clustered and chaotic (Table 6).

**Table 6.** Fractal dimension of the polycentric concept (PC).

| Term | Area/ha/ | Perimeter | Feret's Diameter /Max/ | Feret's Diameter /Min/ | Feret's Diameter (F) | Compactness (C) | Fractal Dimension (D) |
|------|----------|-----------|------------------------|------------------------|----------------------|-----------------|----------------------|
| PC | 39235 | 772190 | 50,321 | 36,932 | 0.73 | 0.09 | 1.23 |

3.2.3. Analysis of the Compact City Concept in Ulaanbaatar City

The compact city is one of the most well recognized sustainable urban forms [1]. In modern times, the concept of a compact city is understood as a planning approach that limits urban sprawl in the face of rapid population growth and leads to a more densely populated urban land-use form [32]. Although the form of a city may vary depending on the characteristics and planning objectives of the city, Danztig and Saati [33] first defined the concept as promoting high urban density, less dependence on cars, and common features of everyday life [34]. In this sense, the key solution to the concept is to reduce urban sprawl by creating a high-density residential area with tall buildings in the readjustment zone of Ulaanbaatar. Land readjustment is a key tool for implementing the compact city concept [35–37]. It not only helps to improve the urban land-use form but also helps landowners to increase the economic efficiency of their land [38]. In addition, land readjustments affect many

fields, such as housing land supply, urban-sprawl prevention, reconstruction after disasters, and readjustment in commercial areas [39]. In the central part of Ulaanbaatar, 1413 buildings were built in an area of 4604.87 ha during the implementation of the monocentric concept. Resolutions 01-01/05 in 2014, 01-01/08 in 2016, and 02-01/03 and 02-01/04 in 2017 formed by the Ulaanbaatar City Specialized Inspection Agency banned the use of 808 apartment buildings that did not meet operational requirements and were not earthquake-resistant in this area. They decided to demolish these buildings and implement land readjustment. Urban sprawl could be reduced by implementing the compact city concept in this area. The average spatial parameters in the current residential area of 9–16 story apartments in Ulaanbaatar were measured to determine the quantitative value of the urban sprawl.

Land-Use Efficiency Analysis

Average data values from 10 densely populated 9–16 story apartment districts in Ulaanbaatar were used to calculate the indicators of land-use efficiency (Table 7).

**Table 7.** Land-use efficiency of the Compact city concept.

| No | Indicator | Unit of Measurement | Before the Project |
|----|-----------|---------------------|--------------------|
| 1 | Building coverage ratio | % | 57.8 |
| 2 | Population density | People per ha | 316 |
| 2 | Floor area ratio | Unit | 4.9 |

Urban land readjustment projects and programs will be key tools for implementing the compact city concept. Therefore, the implementation of the compact city concept in an area of 4604.87 ha for land readjustment and the calculation of the population density data of 316 people/ha yielded the following results regarding the urban carrying capacity of the region (Table 8).

**Table 8.** Estimation of the increase in the urban carrying capacity.

| No | Area | Average Population Density/ before/People/ ha | Average Population Density/After /People/ha | Area Size /ha/ | Population/ Now/ | Population/ after the Project/ | Difference |
|----|------|------|------|------|------|------|------|
| 1 | Land readjustment area | 65.6 | 316 | 4604.87 | 302,079 | 1,455,139 | 1,153,060 |

With the implementation of the compact city concept, a total number of 1,153,060 people will be able to settle in the land readjustment area. Estimation of data in accordance with the general land management plan of Ulaanbaatar showed that this could reduce urban sprawl by 26,756 ha, making the total area 12,479 ha (Figure 8).

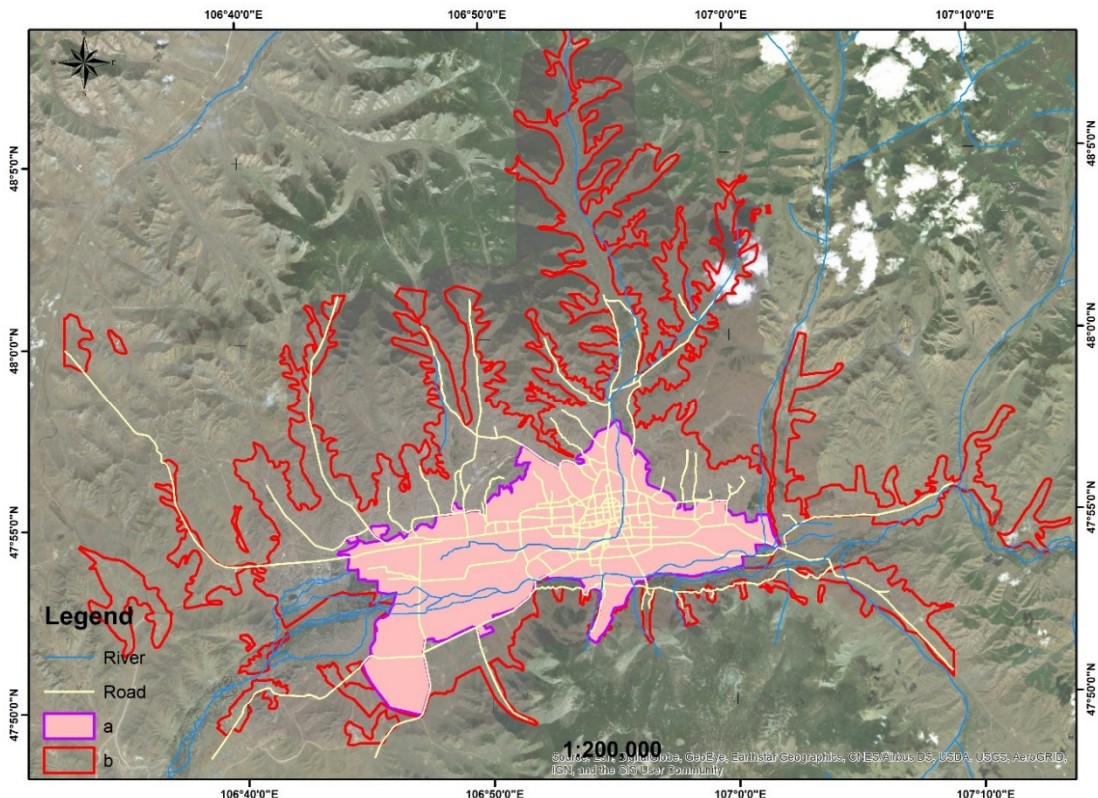

**Figure 8.** Map using the compact city concept: (**a**) After the implementation of the compact city concept; (**b**) urban sprawl by 2020.

Land-Use Form Analysis

The compact city concept is based on the results of an analysis of densely populated, high-rise residential areas, which were used as the above quantitative values. In terms of the compact city concept, as mentioned above, the density of the city center was predicted to quadruple to 0.40, and the area to be built was predicted to reduce to 12,479.7 ha. However, the fact that the fractal dimension, which is an indicator of the urban growth dynamics, is 1.08 shows that the pattern of urban sprawl would improve if this concept was implemented (Table 9).

**Table 9.** Land-use form analysis of the compact city concept (CC).

| Term | Area/ha/ | Perimeter | Feret's Diameter /max/ | Feret's Diameter /min/ | Feret's Diameter (F) | Compactness (C) | Fractal Dimension (D) |
|------|----------|-----------|------------------------|------------------------|----------------------|-----------------|------------------------|
| CC | 12,479 | 97,837 | 22,300 | 13,600 | 0.61 | 0.40 | 1.08 |

### 3.3. Comparison of Urban Planning Concepts

To compare the impact of sprawl in urban planning concepts, it is necessary to combine the corresponding quantitative values of urban land-use form and land-use efficiency during their implementation. The quantitative values related to urban sprawl in the settlement area are as follows: Table 10 shows that the built-up area under the compact city concept would be less than that under the current polycentric concept area by 26,756 ha, and it would be 14,062 ha less than the built-up area under the monocentric concept. Thus, the compact city concept may be effective for reducing urban sprawl in Ulaanbaatar.

### 3.3.1. Land-Use Form Analysis

According to Table 10, Feret's aspect ratio is 0.60 when the monocentric concept is used and 0.61 when the compact city concept is used, both of which elongate the land-use form in the transverse direction. However, when the polycentric concept is used, Feret's aspect ratio is 0.69, which tends to correct the land-use form in the vertical direction. In terms of urban compactness, the population density was shown to be 0.12, that is, sparse, when the monocentric concept was used, but it further decreased to 0.09 when the polycentric concept was used. With the compact city concept, the population density increased to 0.4. Comparing the results for the fractal dimension, for the monocentric and polycentric concepts, the dimensions were 1.21 and 1.23, respectively, resulting in a clustered pattern, while for the compact city concept, the fractal dimension was 1.08, resulting in an improved shape of the city with reduced curves.

### 3.3.2. Land-Use Efficiency Analysis

According to Table 10, the population density, which was 65.6 people per ha when the monocentric concept was used, decreased to 50 people per ha when the polycentric concept was used and increased to 316 people per ha when the compact city concept was used. The floor area ratio was 1.6 for the monocentric concept, it increased slightly to 1.8 with the polycentric concept, and with the compact city concept, the ratio increased by 2.6–2.9 times compared with the other two concepts, up to 4.7. The building coverage ratio of 33.7 with the monocentric concept increased slightly to 43.7 with the polycentric concept, and with the compact city concept, the ratio increased by 1.3 and 1.7 times, respectively, from the other two concepts, up to 57.8.

**Table 10.** Comparison of urban concepts (mc—monocentric, pc—polycentric, cc—compact city).

| Term | Area /ha/ | Perimeter | Feret's Diameter /Max/ | Feret's Diameter/ Min/ | Feret's Diameter (F) | Compactness (C) | Fractal Dimension (D) | Building Coverage Ratio (BCR) | Floor Area Ratio (FAR) | Population Density (r) |
|------|-----------|-----------|------------------------|------------------------|----------------------|-----------------|-----------------------|-------------------------------|------------------------|-----------------------|
| MC | 26,541 | 483,850 | 45,880 | 27,580 | 0.60 | 0.12 | 1.21 | 33.7 | 1.6 | 65.6 |
| PC | 39,235 | 772,190 | 50,321 | 36,932 | 0.73 | 0.09 | 1.23 | 43.7 | 1.8 | 50 |
| CC | 12,479 | 97,837 | 22,300 | 13,600 | 0.61 | 0.40 | 1.08 | 57.8 | 4.7 | 316 |

## 4. Discussion

The study offers a good opportunity to use urban land-use form and land-use efficiency indicators that were used in Houston, Shanghai, and Nanjing cities based on satellite data and GIS technology in a reducing urban sprawl study. According to land-use form analysis, in 1990, Ulaanbaatar city had a corridor urban structure of nine main categories and indicating that the urban sprawl has expanded mostly vertically and has become a dispersed urban structure with low-density sprawl over the last 30 years. However, considering every five years, in 1990–1995 and 1995–2000, the urban sprawl expanded horizontally and became a corridor urban structure, but from 2000 to 2005 it expanded vertically and the structure of the city became relatively standard and compact urban structure. In the following years, 2005–2010, 2010–2015, 2015–2020 the urban sprawl expanded horizontal direction and dispersed urban structure of nine main categories. In the past 30 years, urban density was decreased three times, settlement area of Ulaanbaatar city reached 39,235 ha due to the rapid expansion of low-density sprawl. The low-density sprawl imposes high environmental costs in terms of energy consumption and extra capital costs through the provision of bulk infrastructure and services such as public transport, health, water, and sanitation [4]. Rapid growth in urban populations caused by uncontrolled sprawl created a number of problems related to urban land use [40]. On a global scale, overpopulation and urban development have the ability to increase the occurrence of natural hazards and their impacts both in the developed and developing world [41]. The Ulaanbaatar city land managements' master plan defined the settlement zone area suitable for living as 33,698 ha. These current geospatial technologies are very useful for assessing future hazard occurrences and identifying the vulnerability of communities to hazards [42]. According to land-use form analysis, the settlement zone of Ulaanbaatar city area exceeds the limit by 5537 ha, which is not safe for living. There are 111,086 inhabitants living in 5537 ha of natural hazards area. As for today, urban planning in Ulaanbaatar city has involved a combination of urban concepts, but for future planning, one optimal concept should be identified.

According to the guidance of the Ulaanbaatar city master plan, the indication of the population density should exist 110–460 people per ha and the building coverage ratio is 20–60%. Despite the fact that the floor area ratio has not been indicated in the urban master plan, given the above calculations, the current ratio can be 0.8–5.0. By comparing the research result of the land-use form analysis, according to the three urban concepts, the floor area ratio and the building coverage ratio are within the master plan term limits. The outcomes show that according to the population density, the monocentric concept model is reaching 65.6 people per ha, the polycentric concept model is 50 people per ha, and the compact city model is 316 people per ha. As for the monocentric and polycentric concept models, the above population density is less than the indicated term limits (110–460 people per ha) of the urban master plan which shows the disadvantage of these two concepts. In other words, these two concepts did not decrease the urban sprawl of Ulaanbaatar city. As stated in the research, for the compact city concept model the amount of the population density is exactly fitting into the master plan indicated term limits as well as decreasing the sprawl three times which signifies the advantage of it.

To compare the outcome of the land-use efficiency analysis, according to the three concept models of the Feret's diameter has approximate value, and for the Fractal dimension (D) the monocentric concept model has 1.21, the polycentric concept model has 1.23 and the compact city concept has 1.08 and for the Compactness ratio the concept models of the monocentric equal to 0.12, the polycentric to 0.09 and the compact city to 0.4. Based on the theory of the Fractal dimension the values range between 1 and 2, and if the forms are more complex, the fractal dimension value approaches 2 which indicated the disadvantage of it [21]. Hence, the monocentric and the polycentric concepts are creating a clustered and chaotic form sprawl to Ulaanbaatar city but the compact city concept is more close to 1, which gives the advantage to improve the form of the city. As the theory demonstrates, if the city expands outwards, the compactness ratio decreases, whereas an increased internal density results in an increased compactness ratio [25]. Hence, the monocentric and polycentric concepts in the compactness ratio have decreased with outbound sprawl, nevertheless,

in the compact city concept the compactness ratio has increased which approves the internal density. According to the results of the land-use efficiency analysis, the monocentric and polycentric concepts have a chaotic clustered form with outbound sprawl which causes the disadvantage of it. About the compact city concept, it improves the chaotic clustered form of the Ulaanbaatar city by reducing urban sprawl with increasing internal density, due to which the city form improves and shows its advantage.

To summarize the research result suggests that the compact city concept could be applied to the current circumstances of Ulaanbaatar as it improves urban density and in the meantime reduces urban sprawl threefold and makes the used area of the city more compact. The creation of higher density cities will be more sustainable in the future—both environmentally and financially. Higher density development within the existing urban industry can help to reduce the rate at which peripheral land with agricultural, bio-diversity, and mineral potential is consumed. It might also reduce the need for people to travel to work by car and make more efficient use of the city's existing infrastructure [4]. The research outcome manifests that a comparison of the other two models has a decent result. Both special indicators have been increased by its own, the land-use form analysis with 11%, and land-use efficiency analysis from 2.6 to 2.9%. Therefore, according to the technical point of view, the compact city concept can be the most suitable optimum concept in the current situation of Ulaanbaatar city. The compact city concept has been utilized in the following countries, such as Japan, South Korea, Taiwan, and China, to solve the urban sprawl in their own way [43]. Therefore, land readjustment is the key tool for implementing the compact city concept [35–37]. In addition, land readjustments affect many fields, such as housing land supply, urban-sprawl prevention, reconstruction after disasters, and readjustment in commercial areas [39].

## 5. Conclusions

During 1990–2020, Ulaanbaatar city's residential zone area expanded by 28,505 ha or 3.6 times, and the perimeter increased by 659,590 m or approximately 6.8 times. This shows that the estimated area with favorable living conditions was exceeded by 5537 ha. According to satellite data from 2020, Ulaanbaatar city has expanded vertically and concentrated in different parts, which has caused the urban form to dispersed urban structure.

When the monocentric concept was applied, the urban sprawl was shown to be less compact with partial concentration and it was stretched out horizontally. A less compact city concept is supported by a relatively large area of 26,541 ha. Under this concept, around 70% of the area identified as favorable for suitable settlement would be occupied by a somewhat sparse population. Therefore, there would be limited possibility for settling new residents from the countryside into a suitable settlement zone, which would further serve as a cause of unorganized urban sprawl.

The polycentric concept allows the division of the already formed unrestricted settlement area into subzones and provides these zones with engineering and social infrastructures, hence reducing the pressure imposed on the city center and improving the urban form ratio. However, urban sprawl would continue to be even more intensive and scattered. It is possible that the urban sprawl would be accelerated by enabling unrestricted settlement of engineering infrastructure and social welfare.

In the scope of the compact city concept, it is possible to reduce the current urban sprawl threefold by increasing the density of Ulaanbaatar City's central area, an area of 4604.87 ha. Although the city's current form would be redressed by condensing urban sprawl, this solution does not provide an optimal ratio for urban land-use form. However, the research results suggest that it is possible to reduce urban sprawl if the urban land-use form ratio and building coverage ratio are planned and implemented at the proper rates. This would further allow the population, which has migrated to the capital city in search of better economic opportunities, to settle in a zone with suitable living conditions.

In conclusion, for Ulaanbaatar city, which has a dense population but a small residential area, the use of monocentric or polycentric concepts will not reduce the current urban sprawl or lessen unrestricted sprawl. The compact city concept, on the other hand, could reduce city sprawl by compressing the internal urban form.

**Author Contributions:** Conceptualization, B.B. and C.F.; Data curation, B.M.; Formal analysis, B.M.; Funding acquisition, B.B.; Methodology, B.B. and C.F.; Resources, B.M.; Supervision, C.F.; Writing—original draft, B.B. and B.M.; Writing—review and editing, C.F. All authors have read and agreed to the published version of the manuscript.

**Funding:** This research received no external funding.

**Conflicts of Interest:** The authors declare no conflicts of interest.

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
