# Peer review of "Estimating the Impact of Urban Planning Concepts on Reducing the Urban Sprawl of Ulaanbaatar City Using Certain Spatial Indicators"

_land, doi:10.3390/land9120495_

Round 1
Reviewer 1 Report
The authors made few revisions based on my comments, and should make more efforts to extensively revise this paper (especially in the abstract, introduction, results and discussion parts) before publication.
- The current abstract still provided no meaningful results and policy implications.
- The introduction needs major revision to reflect the related research worldwide and to highlight the importance of your study.
- In the Methods and Data section, the relevant information of multiple data source RS images can be explained more clearly in a self-explanatory table. The introduction of accuracy assessment should be described in more detail, e.g., the number of samples for in each time period, methods of accuracy assessment (confusion matrix?), using a separate paragraph.
- A subsection of accuracy assessment needs be added in the results part.
- More references need to be reviewed and added in the discussion part to reflect the advantages, disadvantages and policy implications of this study.
- The terms of “building coverage” and “building coverage ratio” refer to different meaning? Please use uniform terms. The tables and figures should be self-explanatory, and the abbreviations need be explained in notes.
- The language needs major revision.
Author Response
Dear Reviewer
I revised my article. Please see attachment file.
Best regards
Bolormaa

Reviewer 2 Report
Dear authors,
Thank you for this revised paper. As the paper stands now it has merit for publication in this form if the editors also agree my point of view. The paper was altered in line with my previous comments. All the best and kind regards
Author Response
Dear Reviewer
Thank you for your responce
Best regards
Bolormaa
Reviewer 3 Report
The manuscript entitled “Estimating the Impact of Urban Planning Concepts on Reducing the Urban Sprawl of Ulaanbaatar City Using Certain Spatial Indicators”, by B. Batsuuri, C. Fürst and B. Myagmarsuren, presents an interesting work.
In general, the manuscript should be acceptable for publication but some problems must be repaired prior to publication. Some suggestions are as follows:
1. An extensive discussion would improve your paper.
2. It would be useful to be described the aim of this paper.
3. You could enrich the scientific literature.
4. Please justify convincingly why this manuscript (method, thematology etc) connected with LAND’s content and scope. Perhaps the using of proper literature from this journal would be helpful. Eg:
- Bathrellos, G.D.; Skilodimou, H.D. Land Use Planning for Natural Hazards. Land 2019, 8, 128.
- Skilodimou, H.D.; Bathrellos, G.D.; Koskeridou, E.; Soukis, K.; Rozos, D. Physical and Anthropogenic Factors Related to Landslide Activity in the Northern Peloponnese, Greece. Land 2018, 7, 85.
Author Response
Dear Reviewer
Thank you for your response. I revised my article.
Please see attachment.
Best regards
Bolormaa

Round 2
Reviewer 1 Report
All my comments have been revised, and the quality of this version has been greatly improved and can be published.
Reviewer 3 Report
The manuscript entitled “Estimating the Impact of Urban Planning Concepts on Reducing the Urban Sprawl of Ulaanbaatar City Using Certain Spatial Indicators”, by Batsuuri, C. Fürst and B. Myagmarsuren, presents an improved and good work.
The manuscript should be acceptable for publication in the present form.
This manuscript is a resubmission of an earlier submission. The following is a list of the peer review reports and author responses from that submission.
Round 1
Reviewer 1 Report
The paper used multiple RS images to map the settlement zone area of Ulaanbaatar city, and used three indicators of land use efficiency and three indicators of land use form to estimate the impacts of three urban planning concepts on reducing urban sprawl during 1990-2020. The paper had problems such as chaotic structure, unclear presentation, rough literature review, unclear methods, unreliable results, and meaningless discussions; thus, I think the current paper did not meet the publication requirements.
- The abstract provided excessive background information and method introduction, but did not have meaningful results and policy implications to benefit readers.
- The authors should review more literatures regarding urban planning concepts (monocentric, polycentric and compact city concepts) and their advantages, disadvantages and impacts from an international perspective. The current introduction neither analyzes the global related research progress nor reflects the significance of this research.
- The authors should provide basic information of multiple RS images, such as ID, acquisition time, temporal and spatial resolutions, and download website. The authors need to clarify how to deal with the uncertainty among multiple RS images with different sensors and attributes. How the author did the accuracy assessment for the classification results? How does the author evaluate the impacts of three concepts on reducing urban sprawl?
- The authors should add a detailed accuracy assessment in the results section to increase the credibility of the results.
- The discussion part should strengthen the comparison with other studies to reflect the advantages, disadvantages and policy implications of this study. More literatures need to be reviewed.
- Please change “hectares” to “ha” in whole text, and change “;” to “,” in line 110. Please use uniform terms in whole text, e.g., building coverage and building coverage ratio.
- The language needs major revision.
Reviewer 2 Report
The paper deals with the interesting topic of urban sprawl in relation with the urban restructuring and urban regeneration processes. The paper makes use of relevant urban indicators analysis and represent a current research in the field of urban geographies of Asian cities.
The comments are as follows:
- Abstract
The abstract should be a little bit shortened I think it is too long and it should be more clear and restructured in line with the main objectives, data and methods and the main findings of the article. Please restructure the abstract and design it more clearly.
- The Introduction
This section is well designed but it lacks some major references on the main topic addressed. I suggest to set the main research objective in a general context of urban planning and urban sprawl.
2a. The introduction should be followed by a theoretical background dealing with the concepts of urban sprawl in line with urban planning and also the main concepts and criteria of analysis should be integrated. I suggest to make a new section here with the theoretical background of the main concepts used as well as related to the contemporary vision on urban planning and urban sprawl in line with the present evolution and development of the cities at a global scale. There are a plenty of studies in Web of Science and other data basis dealing with this issues.
2b. In the introduction section I think that the authors have to expand a little the situation of the cities after 1990 in line with the international literature. I think that the book Cities and economic change authored by Paddison R and Hutton P could inspire the authors to improve the introduction.
2c. At the citation [2] in the text (first page) the reference here have to be completed with the following reference titles because this idea appears in these papers: The inclusion of these titles are mandatory here from my side. See below:
JUCU, Ioan Sebastian, (2015), The spatial polarization of small-sized towns in Timiş County of Romania, în SGEM 2015 Conference Proceedings 15th GeoConference on Ecology, Economics, Education and Legislation, Issue 3, pp. 729-736 ISBN 978-619-7105-19-3. DOI: 10.5593/SGEM2015/B53/S21.094, http://sgem.org/sgemlib/spip.php?article6596
- Data and methods are well designed and appropriate used but they need to be closely connected to a theoretical context as I mentioned above.
- Results, discussions and conclusion.
They are clearly provided but also they have to be presented in line with the present theories on urban sprawl and urban planning.
- however the results should be correlated with the theoretical insights which in this paper are not included as theoretical section.
- the spatial analysis have to be more explained in line with the consequences of the city’s urban sprawl and planning actions. This questions have to be more developed.
- the land use rate also has to be interpreting in line with the future actions in the city urban planning.
- the above issues raise because of the absence of an appropriate theoretical background on the current trends at the international level both in urban sprawl and urban planning interventions.
The graphical part of the paper is well prepared and illustrate the content.
English language needs some minor/moderate corrections for clarity.
Reviewer 3 Report
The manuscript entitled “Estimating the Impacts of Urban Planning Concepts on Reducing Urban Sprawl Using Certain Spatial Indicators”, by B. Batsuuri, C. Fürst and B. Myagmarsuren, presents an interesting work.
In general, the manuscript should be acceptable for publication but some serious problems must be repaired prior to publication. It needs some significant improvement. Some suggestions are as follows:
- An extensive discussion would improve your paper.
- Please use different terms in the “Title” and the “Keywords”.
- It would be useful to be described the aim of this paper.
- The English language usage should be checked by a fluent English speaker. It is suggested to the authors to take the assistance of someone with English as mother tongue.
- You could enrich the scientific literature.
- Please justify convincingly why this manuscript (method, thematology etc) connected with LAND’s content and scope. Perhaps the using of proper literature from this journal would be helpful. Eg:
- Bathrellos, G.D.; Skilodimou, H.D. Land Use Planning for Natural Hazards. Land 2019, 8, 128.
- Skilodimou, H.D.; Bathrellos, G.D.; Koskeridou, E.; Soukis, K.; Rozos, D. Physical and Anthropogenic Factors Related to Landslide Activity in the Northern Peloponnese, Greece. Land 2018, 7, 85.
7. The authors could make a discussion about the relationship between urban planning and natural hazards.
8. In all maps you must put coordinates.
9. Correct references in the text and the reference list according to the journal’s format. Please format the references’ list by using the correct journal abbreviations.
See the following link: https://images.webofknowledge.com/images/help/WOS/A_abrvjt.html
10. Please be careful with the spaces between the words.